# Design Optimization of Lattice Structures under Compression: Study of Unit Cell Types and Cell Arrangements

**DOI:** 10.3390/ma15010097

**Published:** 2021-12-23

**Authors:** Kwang-Min Park, Kyung-Sung Min, Young-Sook Roh

**Affiliations:** 1Construction Technology Research Centre, Construction Division, Korea Conformity Laboratories, Seoul 08503, Korea; mks2523@kcl.re.kr; 2Architectural Engineering Program, Department of Architectural Engineering, Seoul National University of Science and Technology, Seoul 01811, Korea; rohys@seoultech.ac.kr

**Keywords:** additive manufacturing, 3D printing, selective laser melting, unit cell, variable-density lattice structures, design optimization, mechanical properties

## Abstract

Additive manufacturing enables innovative structural design for industrial applications, which allows the fabrication of lattice structures with enhanced mechanical properties, including a high strength-to-relative-density ratio. However, to commercialize lattice structures, it is necessary to define the designability of lattice geometries and characterize the associated mechanical responses, including the compressive strength. The objective of this study was to provide an optimized design process for lattice structures and develop a lattice structure characterization database that can be used to differentiate unit cell topologies and guide the unit cell selection for compression-dominated structures. Linear static finite element analysis (FEA), nonlinear FEA, and experimental tests were performed on 11 types of unit cell-based lattice structures with dimensions of 20 mm × 20 mm × 20 mm. Consequently, under the same relative density conditions, simple cubic, octahedron, truncated cube, and truncated octahedron-based lattice structures with a 3 × 3 × 3 array pattern showed the best axial compressive strength properties. Correlations among the unit cell types, lattice structure topologies, relative densities, unit cell array patterns, and mechanical properties were identified, indicating their influence in describing and predicting the behaviors of lattice structures.

## 1. Introduction

### 1.1. Structural Lattice

Cellular structures, such as animal bones, tree trunks, leaves, corals, and honeycombs, exist naturally in the environment [1]. Over the years, human beings have processed various natural cellular structures, such as corks and other wooden products, for various applications, and have manufactured and utilized various artificial cellular structures by imitating natural cellular structures. Polymer-based foam structures, metal-based honeycomb, and truss structures are the commonly used cellular structures [2]. Polymer foams are commonly used in applications such as packaging containers and insulations [3]. Metal honeycomb and truss structures are used as lightweight structures and shock-absorbing materials in various industries, including aerospace, architecture, automobile, and electronic industries [4].

Three-dimensional (3D) cellular structures are generally classified into stochastic (foam) and non-stochastic (periodic) structures depending on the arrangement of their unit cells, and each structure can be further divided into closed-cell and open-cell structures (Figure 1) [5]. Open-cell foam structures have open cell walls, usually created using an inflating agent during production, leaving only the framework in place that allows fluids and gases to easily pass through [6]. In contrast, closed-cell structures have a closed cell wall structure with intact membranes, making the closed-cell foam non-porous and impermeable. Typical stochastic cell structures include metallic and non-metallic foams, which exhibit structures similar to those of sponges [7]. Non-stochastic structures are also known as periodic or lattice structures, in which the unit cells of a specific shape are repeatedly arranged, such as in honeycombs and trusses [8]. 

In particular, lattice structures have received significant attention because of their attractive properties such as high specific strength and stiffness, ultra-low weight, and the ability of their internal structures to absorb an external impact force [9,10,11]. Solid materials (such as steel and plastic) are generally used in industrial structures. Although solid materials meet structural design requirements, many disadvantages persist, such as material wastage during production, inflexible design strategies, and large structure volumes. At present, industrial structural design primarily focuses on reducing the weight and achieving high performance of the products. For example, in aerospace applications [12,13], structural designs aim to achieve excellent mechanical properties using lightweight materials, which allow the structures to be built at a low cost with high performance. The use of lattice structures facilitates the production of structures with reduced weights and high strengths [8].

### 1.2. Additive Manufacturing and Selective Laser Melting

Despite the various advantages of lattice structures, the manufacturing process is complicated because of their complex internal shapes. This has limited the commercialization and wide use of the lattice structures. Three-dimensional lattice structures are manufactured using conventional methods such as casting, weaving, cutting, and joining. However, the restrictions on the design freedom of these methods have limited their further applications [14,15]. 

Additive manufacturing (AM) technologies, which are part of the third manufacturing revolution, have emerged as promising manufacturing methods for lattice structures because of their design freedom. In addition, AM processes enable layer-by-layer construction of materials using a computer-aided design (CAD) model, which allows the fabrication of parts with complex geometries that cannot be achieved using conventional manufacturing processes. The unique feature of AM is its ability to manufacture hollow shapes with complex internal and external geometries using tiny cells known as lattice structures.

According to the ASTM F 2792-12 standard [16], AM processes are classified into seven categories: binder jetting (BJ), direct energy deposition (DED), material extrusion (ME), material jetting (MJ), powder bed fusion (PBF), sheet lamination (SL), and vat photopolymerization (VP). Metal AM has been developed by employing PBF and DED. PBF uses a laser or an electron beam heat source to fuse the bed powder material, whereas DED uses a variable heat source and deposits the solid material directly on the surface. Between the two methods, PBF has the advantage of making complex shapes, such as metal lattice structures, with high precision. The process of PBF begins with the design of a 3D model, which is sliced into several layers. Thereafter, the structure is fabricated by fusing; each layer is sequentially bonded to the adjacent top layer, and a powdered material is spread over the joined layers to enable the processing of the next layer. Hence, the PBF process can be considered a discrete process rather than a continuous one. Depending on the type of power source, PBF can be further classified into two subcategories: laser beam melting (PBF-LB) and electron beam melting (PBF-EB). Representative PBF-LB methods are selective laser sintering (SLS), selective laser melting (SLM), and direct metal laser sintering (DMLS), while the representative PBF-EB method is electron beam melting (EBM). EBM and SLM enable the fabrication of lightweight, highly rigid, fine, and complex lattice structures. The powdered materials commonly used for PBF include stainless steel (e.g., 316L SS) [17], titanium alloys (e.g., Ti-6Al-4V) [18,19,20], and aluminum alloys (e.g., AlSi10Mg) [21,22]. 

### 1.3. Research Objectives

Lattice structures are generally constructed by repeating a unit cell using a certain spatial pattern. Thus, the design of a lattice structure includes the unit cell design and pattern design. A unit cell is the smallest element that can be used to form and characterize the entire lattice structure. When the type of the base material and the relative density are fixed, the mechanical properties of the lattice structures depend mainly on the architecture of the unit cell. Significant efforts have been devoted to finding the optimum cell topology that can provide the best mechanical properties with the least amount of material, maximizing the stiffness/strength-to-weight ratio. Thus, several studies have focused on the role of cell topology in enhancing the mechanical properties of the fabricated materials [23,24].

Consequently, understanding the material topology–property relationship, which presents the mechanical properties as a function of the relative density and topology of lattices, is a precondition to allowing a product design that provides improved properties. Therefore, it is necessary to optimize the lattice structure configuration, unit cell array, and unit cell type under axial compressive loading conditions and perform finite element analysis (FEA) and experimental tests on metallic specimens generated by SLM. Therefore, in this study, we addressed the aforementioned issues by implementing the following processes, as shown in Figure 2.

We aimed to develop an approach for the configuration optimization of the lattice structure. AM increases the design freedom, and it remains difficult to design or select an appropriate unit cell topology. In this study, a ground structure topology optimization approach was developed for the unit cell design. Thereafter, we intended to construct and provide a unit cell relative density database according to the box size and circular cross strut radius of the boundary unit cell;We aimed to optimize the unit cell arrangement. We modeled the lattice structures by patterning 1 × 1 × 1, 2 × 2 × 2, 3 × 3 × 3, and 4 × 4 × 4 unit cells to derive the yield strength in order to identify the mechanical properties of the lattice structures. In this study, we only focused on lattice structures with dimensions of 20 mm × 20 mm × 20 mm;We aimed to derive an optimal lattice structure consisting of the optimal unit cell obtained using the abovementioned steps under axial compressive loading conditions because the choice of the appropriate lattice topology is a major challenge in lattice adoption.

## 2. Materials and Methods

### 2.1. Lattice Configuration

#### 2.1.1. Lattice Pattern Setting

The mechanical properties of lattice structures can be predicted as a function of their unit cell geometries, relative densities, sizes, strut dimensions, and arrangement. In particular, for lattice structures, relative density, which is the ratio of the apparent density of the lattice structure to the density of the material with a solid structure, dictates its mechanical properties [25,26,27,28]. 

In this study, 11 types of unit cells were selected via literature review, which were simple cubic, body-centered cubic, face-centered cubic, body center, diamond, truncated cube, truncated octahedron (a.k.a. Kelvin), octahedron, rhombicuboctahedron, octet-cross (a.k.a. octet-truss), and cuboctahedron (Table 1).

#### 2.1.2. Lattice Parameter Setting

The parameters for the lattice structure design were described below, and the ratio of the cross-sectional circular radius to the length of the cube edge (r/s) was derived for each unit cell.

Lattice-structured cubes with nominal dimensions of 20 mm × 20 mm × 20 mm were designed. A specific unit cell was then designed in the cube (s=20 mm);The unit cell strut was created in a circular cross-section to generate a volume in the lattice structure. Specific relative density was designed by controlling r (Figure 3).

#### 2.1.3. Lattice Layout Setting

To fabricate the lattice structure, the unit cells were arranged in a 3D space by repeating the lattice points, which is referred to as the pattern design. A lattice structure can be created from an array of repeated unit cells using direct patterning, in which the unit cells are directly generated by repeating the unit cells in three dimensions (along the *x*-, *y*-, and *z*-axis). For example, a 3 × 3 × 3 lattice structure can be constructed by repeating the unit cell translationally three times in each coordinate axis (Figure 3).

To automate the process, we developed a lattice structure generator plugin for Rhinoceros (Rhino 7, Robert McNeel & Associates, Seattle, WA, USA). This plugin has various unit cell topologies, and it can adjust the lattice structure boundary size, unit cell arrangement, and strut radius of the unit cell (Figure 4).

### 2.2. Optimizing Unit Cell Array

#### 2.2.1. Numerical Modeling

Numerical modeling of the optimal arrangement of unit cells for cubic lattice structure was performed to investigate the axial compressive behavior. The dimensions of the cubic lattice structure were designed to be 20 mm × 20 mm × 20 mm, referring to the standard ISO 13314 [74] and related studies [75,76,77]. The selected unit cell types were simple cubic, body-centered cubic, and face-centered cubic, and their relative densities were designed as 0.1, 0.2, and 0.3, respectively, considering the lightweight structure. The lattice structures were modeled by patterning the unit cells in 1 × 1 × 1, 2 × 2 × 2, 3 × 3 × 3, and 4 × 4 × 4 units to derive the maximum applied compression force at the yield stress (hereinafter referred to as “yield force”) for determining the mechanical properties of the lattice structures. The final lattice structure was created by merging the arrangement interfaces of the unit cells.

#### 2.2.2. Numerical Simulation

FEA was performed using SolidWorks 2018 (Dassault Systems, Paris, France) to determine the optimal arrangement of lattice structures for the axial compressive load. The values of the material properties of the titanium alloy (Ti-6Al-4V) used in this study were derived experimentally. The specimen was manufactured following the ASTM B988-18 standard [78], and the manufacturing parameters were the same as those mentioned in Section 2.3.2. In terms of the results, an elastic modulus of 119.0 GPa, yield strength of 1125.0 MPa, tensile strength of 1200.0 MPa, and Poisson’s ratio of 0.34 were achieved. The derived mechanical properties satisfied the ASTM B988-18 standard (yield strength ≥ 828 MPa and ultimate tensile strength ≥ 895 MPa). The aforementioned material property values were used for the numerical analysis.

The structure has an irregular shape. A free mesh was applied using second-order tetrahedral elements. Considering the convergence, the size of the mesh element was set to 0.6705 mm. The total number of meshes varied depending upon the lattice structure (relative density of 0.3 and 3 × 3 × 3 pattern) types, and the total elements ranged from 57,512 to 100,727 (Table 2). The mesh was refined at the expected point of maximum stress.

A fixed-boundary condition was applied to the bottom surface of the structure; meanwhile, a compression force with a direction orthogonal to the top surface was applied to the top surface (Figure 5). To perform the analysis, the Fourier finite-element plus (FFEPLUS) iterative solver was applied. The FFEPLUS employs advanced matrix reordering techniques that are approximate, and, in each iteration, a solution is assumed, and iterations continue until the error becomes acceptable. The solver either utilizes a maximum number of iterations or a stopping threshold to determine the convergence between consecutive iterations. In this study, maximum iteration and threshold values of 1,000,000 and 0.0001, respectively, were used.

Linear and nonlinear analyses were performed in this study. Upon considering the computational cost, initially, linear analysis was performed on different types of lattice structures to examine the stress trends. The maximum compression force was achieved at the point wherein the von Mises stress met the yield stress of the material. Meanwhile, four types of lattice structures, having the highest compression forces, were selected for nonlinear analysis. During nonlinear analysis, the maximum compression force was achieved at the point wherein the principal stress met the yield stress of the material. All the stress values were calculated based on the elemental mean values.

### 2.3. Optimization of Unit Cell Type under Axial Compressive Loading Condition

#### 2.3.1. Numerical Modeling

Numerical modeling of the optimal unit cell type for cubic lattice structure cubes (20 mm × 20 mm × 20 mm) was performed to investigate the axial compressive behavior. The relative density was fixed at 0.3. The lattice structures were designed with 11 types of unit cells by applying the optimal arrangement obtained in Section 2.2. Through the linear static FEA, the optimal unit cell type was selected from the top four, and the mechanical load test was performed. Here, the linear static FEA was performed using the same method as in Section 2.2. 

#### 2.3.2. Experimental Tests

Lattice structures were manufactured using SLM (Metal3D Metalsys 250E, Ulsan, Korea). In this study, spherical Ti-6Al-4V Grade 5 titanium powder (chemical composition according to ASTM F 2924 [79]) with particle sizes ranging from 10 to 45 μm was used. The production was performed in an inert atmosphere, and the samples were built on top of a solid titanium substrate. The thickness of the layer was 0.02 mm, and the following parameters were applied: laser power of 185 W, scanning speed of 1100 mm·s^−1^, and hatching distance of 90 µ; these parameters were kept constant for all specimens. The building volume of the manufacturing equipment was 250 mm × 250 mm × 250 mm. The specimens were placed on the same layer of the building volume and manufactured in batches, and the manufacturing time was approximately 20 h. 

The compression test was performed following the ISO standard (ISO 13314 [74]), which is the standard testing method for porous and cellular metals. A universal testing machine (UTM, Instron Universal Testing Machine, Norwood, Massachusetts, USA) was used, with a maximum loading capacity of 100 kN. The force was applied at a compression speed of 0.01 mm/s, and the deformation in the sample was measured in the vertical direction. The load was removed after the yield stress was reached. 

## 3. Results and Discussion

### 3.1. Lattice Configuration

The relative densities and r/s values of the 11 types of unit cells (relative density in the range of 0.005–0.900) are shown in Table 3 and Figure 6. By setting r/s as a variable, it was possible to control the relative densities of the unit cells. For unit cells with the same relative density, the r/s of the simple cubic was the largest. At a relative density of 0.3, simple cubic had the highest r/s value, and octet-cross had the lowest value, at 0.206 and 0.087, respectively. It was impossible to formulate a design with high r/s values for the unit cell types of cuboctahedron, face-centered cubic, rhombicuboctahedron, and octet-cross. This was due to a Boolean operation error, whereby an increment in the r/s value resulted in an increase in the volume of the members (node and strut), thereby creating additional intersections between the adjacent members and too many overlapping members.

### 3.2. Optimizing Unit Cell Array

The strut-type unit cell topology can be characterized by the Maxwell number, *M*, which is dependent on the number of struts ($N_{s})$ and nodes ($N_{n})$ (Equation (1)) [71]. The Maxwell values of the unit cell used in this study are shown in Table 4. The number of nodes and struts in each unit cell was derived, and the Maxwell stability was calculated according to Equation (1).
(1)$$M=N_{s}-3N_{n}+6$$

The design is simple, as the smallest strut number is 8 for the body center. For the rhombicuboctahedron, the value is 48. Hence, the design is complicated. Furthermore, the Maxwell stability is the minimum for the truncated cube and truncated octahedron at −30, and the maximum for the face-centered cubic and octet-cross at 0. 

If *M* < 0, there are very few struts to equilibrate external forces without equilibrating moments induced at the nodes, causing bending stresses to develop in struts and leading to a bending-dominated behavior. However, if *M* ≥ 0, external loads are equilibrated by the axial tension and compression in struts, implying that no bending occurs at the nodes, which makes these structures stretch-dominated [44]. Because of these phenomena, stretch-dominated structures are stiff and strong, especially considering their mass, whereas bending-dominated structures are compliant and deform more consistently [80]. All unit cells in this study showed Maxwell stability values equal to or less than 0. Therefore, they are structures that are dominant in bending and stable in axial stretching.

Table 5 and Figure 7 show the yield force by a linear static analysis of the lattice structures at different relative densities (0.1, 0.2, and 0.3) with different unit cell arrangements (1 × 1 × 1, 2 × 2 × 2, 3 × 3 × 3, and 4 × 4 × 4). In lattice structures with the same unit cell type and arrangement, an increase in the relative density significantly increased the yield force of the structure. For example, for the 1 × 1 × 1 simple cubic structure, at the relative densities of 0.1, 0.2, and 0.3, the yield forces of the lattice structures were 3070 N, 8400 N, and 14,470 N, respectively. By increasing the relative density from 0.1 to 0.2 and from 0.1 to 0.3, the yield force increased by 174% and 371%, respectively.

Under the same relative density and cell arrangement conditions, the yield force of the structures decreased in the order of simple cubic, body-centered cubic, and face-centered cubic. 

For a simple cubic system with a relative density of 0.1, the yield forces of the 1 × 1 × 1, 2 × 2 × 2, 3 × 3 × 3, and 4 × 4 × 4 lattice structures were 3070 N, 3715 N, 4890 N, and 4860 N, respectively, i.e., the yield forces increased gradually (in some cases, the yield force decreased when moved from the 3 × 3 × 3 system to the 4 × 4 × 4 system). At a relative density of 0.2, the yield forces of the 1 × 1 × 1, 2 × 2 × 2, 3 × 3 × 3, and 4 × 4 × 4 lattice structures were 8400 N, 13,200 N, 14,880 N, and 30,160 N, respectively. Similarly, at a relative density of 0.3, the yield forces of the 1 × 1 × 1, 2 × 2 × 2, 3 × 3 × 3, and 4 × 4 × 4 lattice structures were 14,470 N, 22,220 N, 30,160 N, and 31,315 N, respectively. These results indicate that, at a constant relative density, the yield strength can be optimized by configuring the lattice structure with different pattern designs. As a result, the yield force recorded for the 3 × 3 × 3 pattern showed a convergence tendency, and excellent compressive strength was observed for this system. 

To further investigate the stress concentration area of the lattice structures, linear static FEA was conducted to calculate the stress distribution of lattice structures under an axial compression load. Figure 8 shows the stress distribution of the titanium alloy lattice structures with a relative density of 0.3 and 3 × 3 × 3 pattern. The simple cubic and face-centered cubic structures caused stress concentration at the element edge. In contrast, in the body-centered cubic structure, stress concentration occurred at the center of the element. This indicated that the introduction of the optimized radius released the stress concentration of the lattice structure at the element edge, improved the compressive capacity, and improved the mechanical properties at the same relative density.

In addition, we performed preliminary experiments to improve the design of simple cubic systems by changing the fillet function (Figure 9). While using the same material, dimensions and relative density, the preliminary results showed that the topology of the edge conditions affected the yield force, modeled by the linear static FEA. According to the linear static FEA results, the stress concentration occurred at the edge. Therefore, the maximum yield force was expected to be improved through the shape-optimized design of rounding-edged simple cubic. As shown in Figure 9, the fillet was created by adjusting the corner radius, and the strut thickness was readjusted to match the relative density. As a result, the yield force was improved to 17,780 N through shape optimization, compared to the basic model, which had a yield force of 14,470 N. The possibility of improving the mechanical performance by edge shape optimization can be explored further in future studies.

### 3.3. Optimization of Unit Cell Type under Axial Compressive Loading Condition

FEA results are shown in Figure 10. The top four types of lattice structures with excellent performance under the axial compression condition were simple cubic, octahedron, truncated cube, and truncated octahedron. 

The stress concentration areas of the four lattice structures as a result of the linear static FEA are shown in Figure 11, except those of the simple cubic, which are shown in Figure 8. All lattice structures caused stress concentrations at the element edge areas. In the case of the octahedron lattice structure, the stress concentration was at the bottom of the structure. 

Furthermore, experimental axial load test results of these four cell types are shown in Figure 12 and Figure 13. In addition, nonlinear FEA was performed on these four models, and the corresponding yield forces are listed in Table 5.

Cell types with yield forces from the highest to the lowest were simple cubic, octahedron, truncated octahedron, and truncated cube, and the yield forces were 86,300 N, 78,470 N, 65,120 N, and 63,680 N, respectively (Table 6). 

In the cases of simple cubic and truncated cube, the initial fracture occurred at the node edge of the upper part of the lattice structure (Figure 11, prior to collapse, yellow dotted circles), which corresponded to the stress concentrated areas as per the linear static FEA results (Figure 8 and Figure 11). According to the progressive collapse in Figure 11, the vertical member at the top of the lattice structure has undergone buckling and crushing [81]. In the case of the truncated octahedron, the initial fracture occurred at the node edge of the upper part of the lattice structure, and it is at the same region where the stress concentration occurred as per the linear static FEA results (Figure 11). According to the progressed collapse images shown in Figure 11, the vertical members at the top of the lattice structure were diagonally sheared [81]. In the case of an octahedron, an initial fracture occurred at the node edge of the bottom of the lattice structure, which is the same position where the stress concentration occurred, according to the FEA results (Figure 11). The diagonal members at the bottom of the lattice structure were crushed. Hence, the compressive deformation and fracture mechanism are highly dependent on the unit cell and lattice topology. The deformation and fracture modes can be predicted through the stress concentration modeled by FEA. Among all the lattice structures examined, the simple cubic and octahedron demonstrated the best mechanical properties under axial compressive conditions. 

The linear static FEA yield forces were 34.9%, 32.5%, 30.0%, and 31.9% of the experimental yield forces for the simple cubic, truncated cube, truncated octahedron, and octahedron systems, respectively. In addition, the nonlinear FEA yield forces were 75.6%, 78.1%, 72.7%, and 72.4% of the experimental yield forces for simple cubic, truncated cube, truncated octahedron, and octahedron, respectively. As a result, the nonlinear FEA showed relatively higher reliability than the linear static FEA, which was approximately 72.4% to 78.1% of the experimental test results. Although the linear static FEA was approximately 30.0% to 34.9% of the experimental test results, it was sufficient to predict the compression fractures according to the stress concentrations.

## 4. Conclusions

Our investigation revealed that the unit cell topology, relative density, and pattern arrangement played an important role in determining the mechanical properties of the additive manufactured lattice structures. The lattice structure design was established, and the mechanical properties under axial compressive load conditions were evaluated through FEA and experimental tests. Consequently, under the same relative density conditions, it was confirmed that the simple cubic lattice structure with a 3 × 3 × 3 array pattern had the best axial compressive strength properties among all the analyzed lattice structures. The main findings of this study are summarized as follows:1.The unit cell topologies of the lattice structures were designed as described in Section 3.1 by determining r/s ratios. It was possible to control the relative density of the unit cells using the r/s datasheet. The lattice structure was created from an array of repeated unit cells using direct patterning, in which the unit cells were directly generated by repeating the unit cells in three dimensions;2.Under the same unit cell type and arrangement conditions, an increase in the relative density significantly enhanced the yield force of the lattice structure;3.The yield force of the lattice structure was optimal for the 3 × 3 × 3 pattern. These results indicated that, at a constant relative density, the yield force could be optimized by configuring the lattice structure through the pattern design;4.Simple cubic, octahedron, truncated cube, and truncated octahedron lattice structures exhibited higher axial compression yield forces compared with body-centered cubic, face-centered cubic, body center, diamond, rhombicuboctahedron, octet-cross, and cuboctahedron structures with the same relative density, indicating that these four lattice structures had better mechanical properties under axial compression conditions;5.Under a compressive load, the initial fracture of the lattice structures occurred at the locations where stress was concentrated. Therefore, when nodes and struts were introduced and under conditions of fillet optimization, the stress concentration were released and re-distributed. At the same time, the compressive strength increased with the optimized node, strut, and fillet design.

Through FEA, the compressive strength characteristics and fracture mode of the lattice structure can be predicted. The compressive deformation and fracture mechanism are highly dependent on the unit cell and lattice topology, and therefore strut edge optimization using unit cell and lattice topology needs to be explored. In the future, we plan to continue our research by adding a create edge filet function (e.g., smooth pipe function) to the lattice structure generator plugin for Rhinoceros. We plan to design a lattice structure using shape-optimized unit cells and study the optimal unit cells and optimal arrangement.

This study highlights that the FEA of lattice structures is dependent on the material model, and practical experimental tests are necessary to obtain reliable results. The FEA methodology developed in this study could provide a foundation for accelerating the development and adoption of lattice structures. This study was conducted using a limited structure size (dimensions of 20 mm × 20 mm × 20 mm). Therefore, it is essential to analyze and optimize the actual size to be manufactured. Moreover, further investigation is necessary to confirm this behavior in parts with larger overall dimensions. In addition, the unit cell types, lattice structure topologies, relative densities, and unit cell array patterns should be determined with respect to the actual structural requirements (such as compression member, bending member, and energy absorbing member).

This study proposes the novelty design guide of variable-density in each unit cell using the ratio of cross-sectional circular radius to the length of the cube edge (r/s) as a variable. In addition, this study suggests a design process guide for an optimized lattice structure based on controlling the unit cell type and cell arrangement, and it shows an optimal lattice structure design possibility under conditions of same relative density.

## Figures and Tables

**Figure 1 materials-15-00097-f001:**
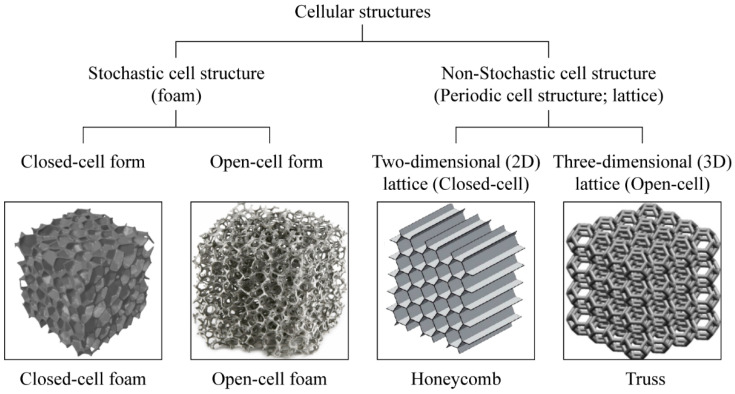
Categories of cellular structures.

**Figure 2 materials-15-00097-f002:**
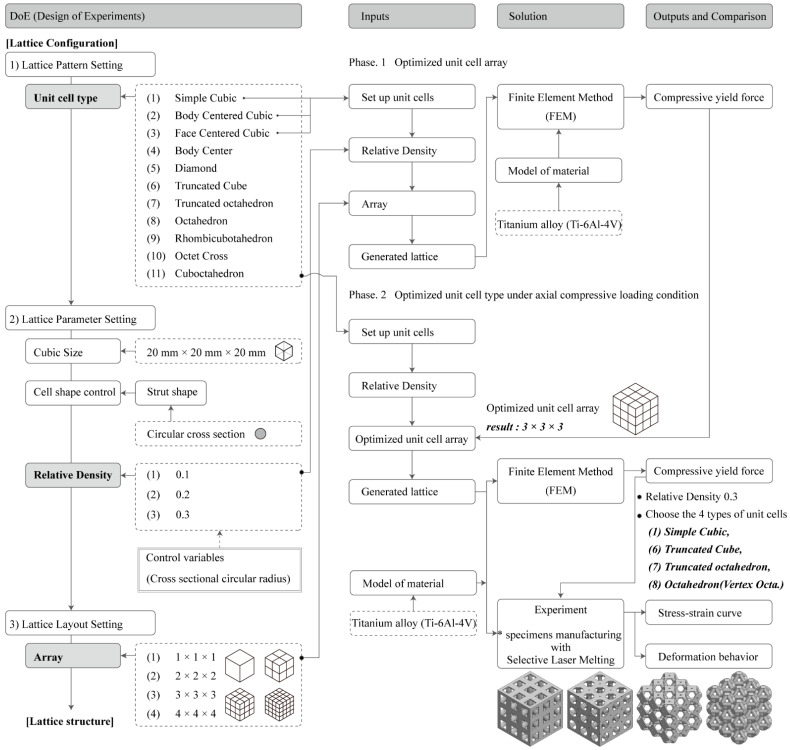
Lattice configuration optimization method and scheme of the study.

**Figure 3 materials-15-00097-f003:**
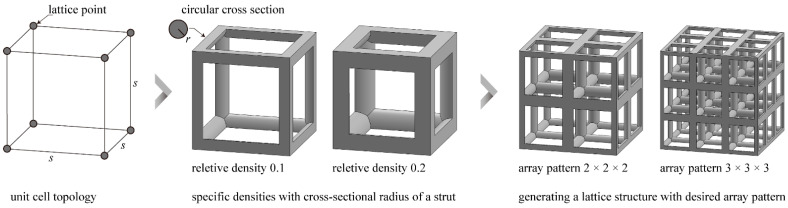
Scheme of the lattice configuration optimization method.

**Figure 4 materials-15-00097-f004:**
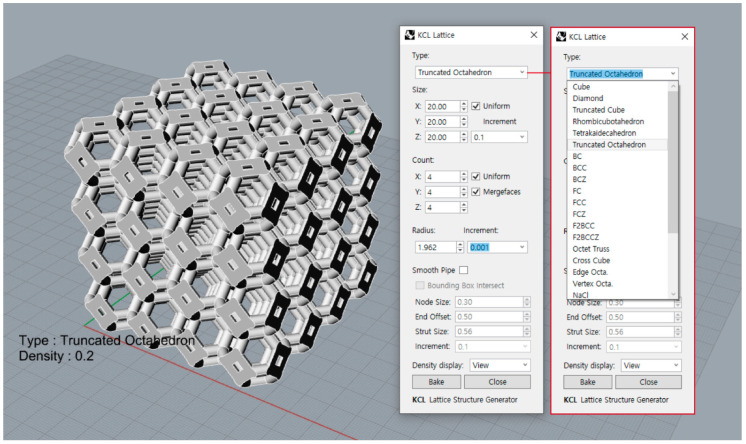
Screenshot of the lattice structure generator plugin for Rhinoceros.

**Figure 5 materials-15-00097-f005:**
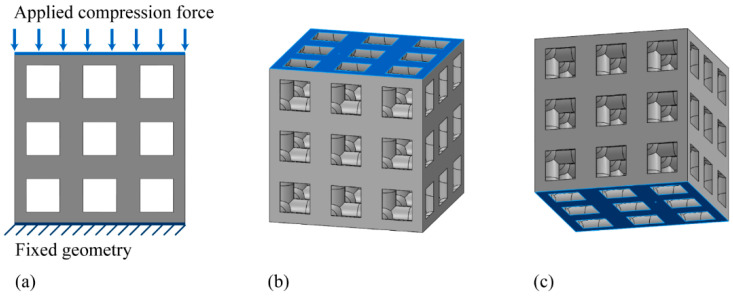
Boundary conditions: (**a**) frontal view with boundary conditions; (**b**) prospective view with applied compression force on top surface; (**c**) prospective view with fixed bottom surface.

**Figure 6 materials-15-00097-f006:**
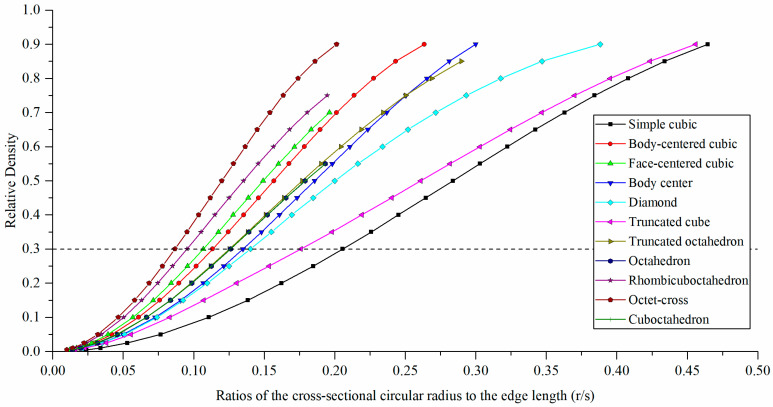
Plot of relative densities vs. ratios of the cross-sectional circular radius to the edge length.

**Figure 7 materials-15-00097-f007:**
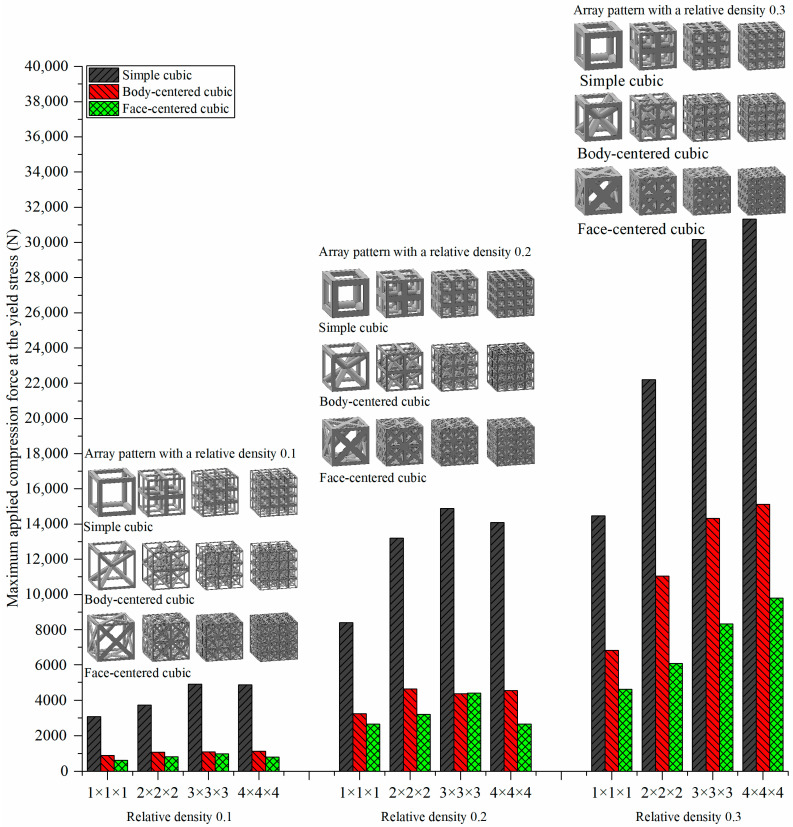
Yield force of the lattice structures with different relative densities and cell arrangements.

**Figure 8 materials-15-00097-f008:**
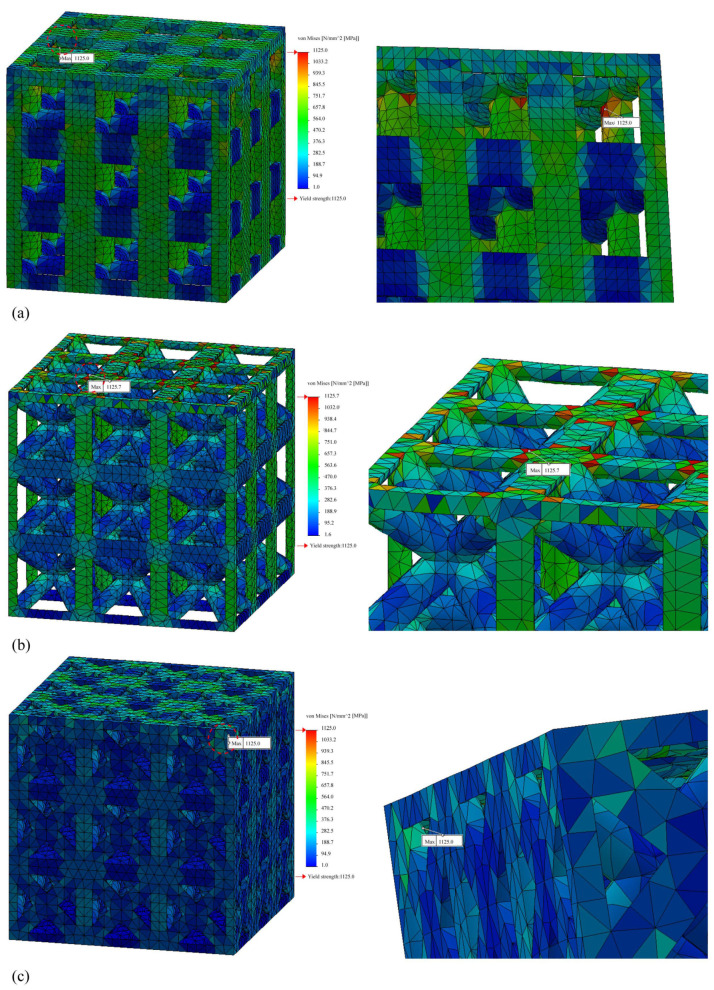
Stress distribution of the lattice structures (relative density of 0.3; 3 × 3 × 3 pattern): (**a**) Simple cubic; (**b**) body-centered cubic; (**c**) face-centered cubic.

**Figure 9 materials-15-00097-f009:**
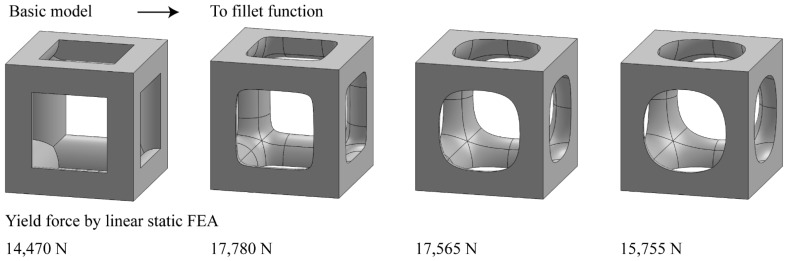
Yield force according to fillet function (simple cubic with a relative density of 0.3; titanium alloy cubic structure of dimensions 20 mm × 20 mm × 20 mm).

**Figure 10 materials-15-00097-f010:**
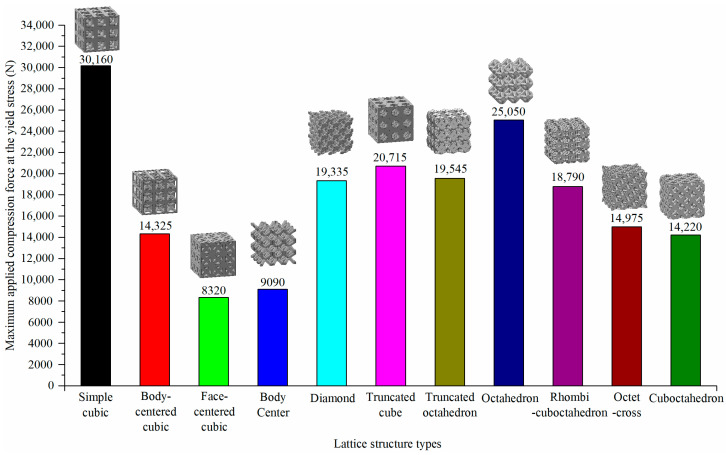
Yield force of different lattice structure types with a relative density of 0.3 and 3 × 3 × 3 pattern.

**Figure 11 materials-15-00097-f011:**
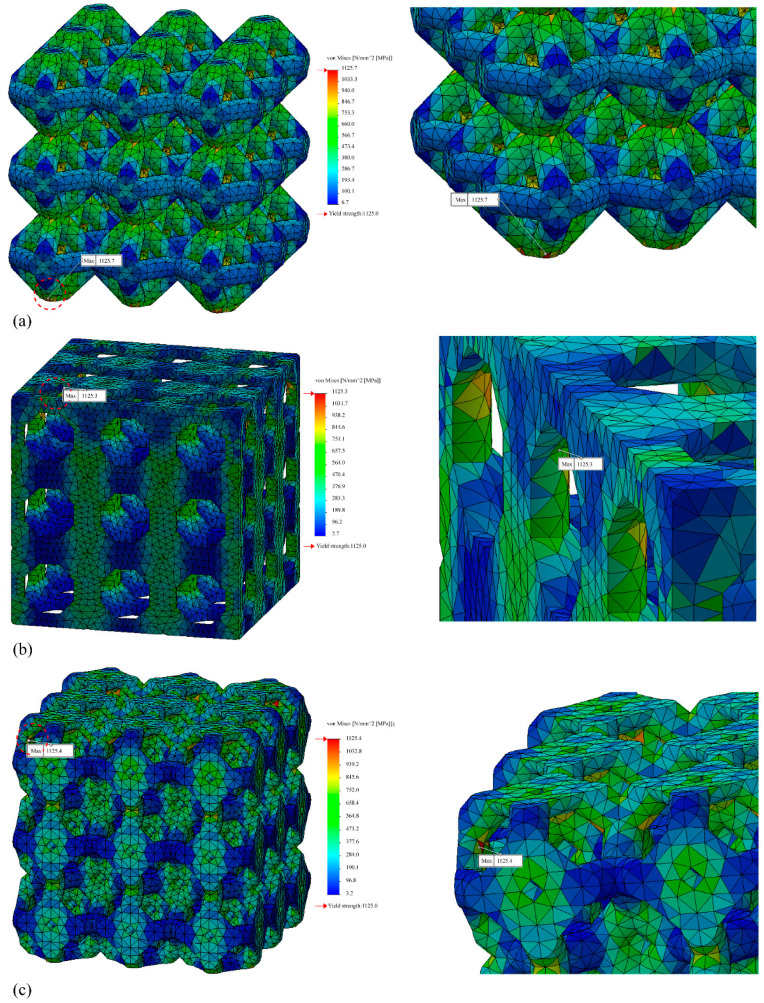
Stress distribution of the lattice structures (relative density of 0.3; 3 × 3 × 3 pattern): (**a**) Octahedron; (**b**) truncated cube; (**c**) truncated octahedron.

**Figure 12 materials-15-00097-f012:**
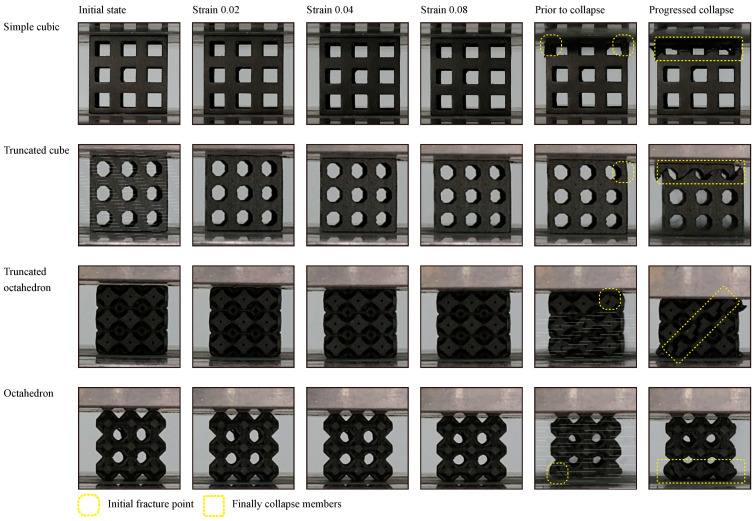
Compressive deformation response of lattice structures with relative density of 0.3 and 3 × 3 × 3 pattern (simple cubic, truncated cube, truncated octahedron, and octahedron).

**Figure 13 materials-15-00097-f013:**
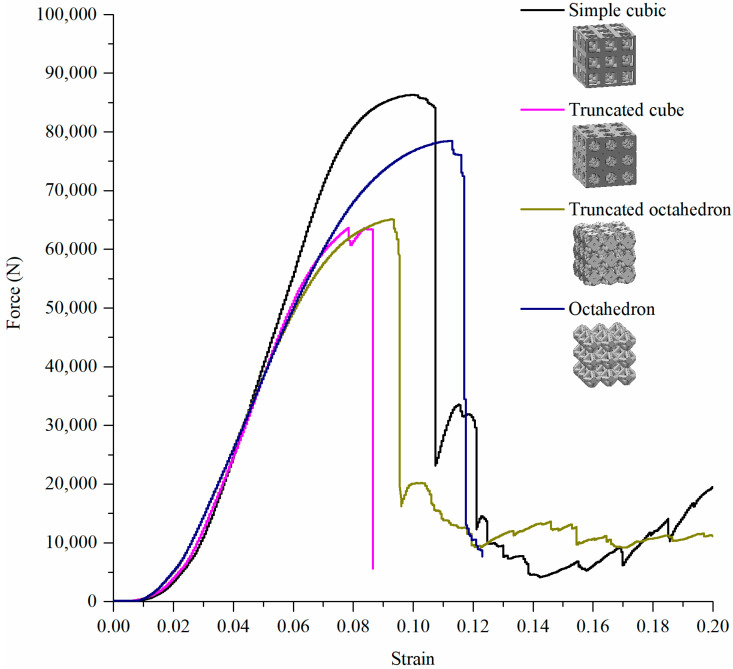
Force–strain response of lattice structures at a relative density of 0.3 and for the 3 × 3 × 3 pattern (simple cubic, truncated cube, truncated octahedron, and octahedron).

**Table 1 materials-15-00097-t001:** Unit cell types considered for the lattice generation.

Unit Cell Topology	Image ^1^	Unit Cell Topology	Image ^1^
Simple cubic [29,30,31,32,33,34,35,36,37,38,39]	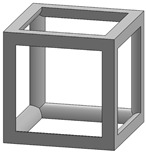	Body-centered cubic [40,41,42]	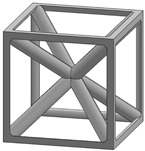
Face-centered cubic [43,44,45]	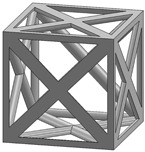	Body center [32,43,44,46,47,48,49,50,51,52,53,54,55,56,57,58,59]	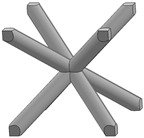
Diamond [29,30,33,34,60,61,62,63,64]	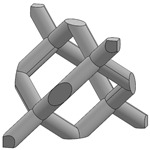	Truncated cube [29,65]	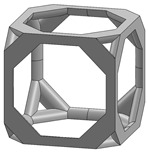
Truncated octahedron (a.k.a. Kelvin) [30,33,34,65,66]	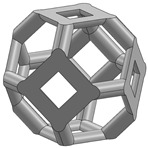	Octahedron [65,67]	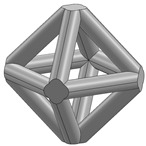
Rhombicuboctahedron [29,65,68]	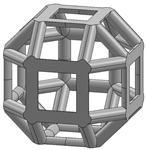	Octet-cross (a.k.a. octet-truss) [69,70,71,72,73]	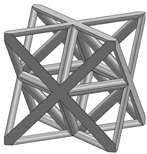
Cuboctahedron [29,65]	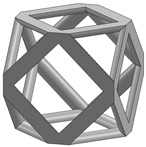		

^1^ Images in this table demonstrate a relative density of 0.1.

**Table 2 materials-15-00097-t002:** Mesh details of the different lattice structures (relative density of 0.3; 3 × 3 × 3 pattern).

	Simple Cubic	Body-Centered Cubic	Face-Centered Cubic	Body Center	Diamond	Truncated Cube	Truncated Octahedron	Octahedron	Rhombicuboctahedron	Octet-cross	Cuboctahedron
Total nodes	97,644	136,270	137,365	105,205	131,167	155,709	145,548	159,411	166,543,	179,549	119,672
Total elements	57,512	77,665	77,719	60,962	78,211	96,331	83,908	98,685	93,887	100,727	68,796

**Table 3 materials-15-00097-t003:** Relative densities of unit cells determined by ratios of the cross-sectional circular radius to the edge length (r/s ).

Relative Density	r/sValues										
	Simple Cubic	Body-centered Cubic	Face-Centered Cubic	Body Center	Diamond	Truncated cube	Truncated Octahedron	Octahedron	Rhombicuboctahedron	Octet-cross	Cuboctahedron
0.005	0.024	0.013	0.012	0.015	0.015	0.016	0.014	0.014	0.010	0.010	0.015
0.010	0.034	0.018	0.017	0.022	0.022	0.023	0.020	0.020	0.015	0.014	0.020
0.025	0.053	0.030	0.028	0.035	0.035	0.038	0.032	0.032	0.024	0.022	0.032
0.050	0.077	0.042	0.039	0.050	0.051	0.056	0.046	0.046	0.035	0.032	0.046
0.100	0.111	0.061	0.057	0.073	0.074	0.083	0.067	0.067	0.050	0.047	0.067
0.150	0.138	0.076	0.071	0.091	0.093	0.107	0.083	0.084	0.063	0.058	0.084
0.200	0.162	0.090	0.084	0.107	0.110	0.131	0.098	0.099	0.075	0.068	0.099
0.250	0.185	0.102	0.096	0.121	0.125	0.154	0.112	0.113	0.085	0.078	0.113
0.300	0.206	0.113	0.107	0.135	0.140	0.176	0.125	0.126	0.095	0.087	0.126
0.350	0.226	0.125	0.118	0.148	0.155	0.198	0.138	0.139	0.105	0.095	0.139
0.400	0.245	0.135	0.128	0.161	0.170	0.219	0.151	0.152	0.115	0.104	0.152
0.450	0.265	0.146	0.139	0.173	0.185	0.240	0.164	0.166	0.125	0.112	0.165
0.500	0.284	0.157	0.149	0.186	0.200	0.261	0.177	0.179	0.135	0.120	0.179
0.550	0.303	0.168	0.160	0.198	0.216	0.282	0.190	0.193	0.146	0.128	0.193
0.600	0.322	0.179	0.171	0.211	0.234	0.303	0.204	N/A ^1^	0.157	0.137	N/A
0.650	0.342	0.190	0.183	0.224	0.252	0.325	0.219	N/A	0.168	0.145	N/A
0.700	0.363	0.201	0.196	0.237	0.272	0.347	0.234	N/A	0.181	0.154	N/A
0.750	0.384	0.214	N/A	0.251	0.293	0.370	0.250	N/A	0.195	0.164	N/A
0.800	0.408	0.228	N/A	0.265	0.318	0.395	0.269	N/A	N/A	0.174	N/A
0.850	0.434	0.243	N/A	0.281	0.347	0.424	0.290	N/A	N/A	0.186	N/A
0.900	0.465	0.264	N/A	0.300	0.388	0.456	N/A	N/A	N/A	0.201	N/A

^1^ N/A: not able to design.

**Table 4 materials-15-00097-t004:** Maxwell stability (*M*) of different unit cell types.

	Simple Cubic	Body-Centered Cubic	Face-Centered Cubic	Body Center	Diamond	Truncated cube	Truncated Octahedron	Octahedron	Rhombicuboctahedron	Octet-cross	Cuboctahedron
No. of nodes	8	9	14	9	14	24	24	6	24	14	14
No. of struts	12	20	36	8	12	36	36	12	48	36	24
M-value	−6	−1	0	−13	−24	−30	−30	0	−18	0	−6

**Table 5 materials-15-00097-t005:** Lattice structure yield forces according to relative density and arrangement.

Relative Density	Lattice Structure Yield Forces (N)											
	Simple Cubic				Body-Centered Cubic				Face-Centered Cubic			
	1 × 1 × 1	2 × 2 × 2	3 × 3 × 3	4 × 4 × 4	1 × 1 × 1	2 × 2 × 2	3 × 3 × 3	4 × 4 × 4	1 × 1 × 1	2 × 2 × 2	3 × 3 × 3	4 × 4 × 4
0.1	3070	3715	4890	4860	890	1065	1095	1135	615	815	975	810
0.2	8400	13,200	14,880	14,095	3225	4630	4350	4540	2650	3195	4385	2655
0.3	14,470	22,200	30,160	31,315	6825	11,045	14,325	15,115	4605	6080	8320	9800

**Table 6 materials-15-00097-t006:** Comparison of FEA and experimental results.

	Simple Cubic	Truncated Cube	Truncated Octahedron	Octahedron
Yield force by experimental test (N)	86,300	63,680	65,120	78,470
Yield force by linear static FEA (N)	30,160	20,715	19,545	25,050
Linear static FEA/experimental test (%)	34.9	32.5	30.0	31.9
Yield force by nonlinear FEA (N)	65,250	49,750	47,350	56,800
Nonlinear FEA/experimental test (%)	75.6	78.1	72.7	72.4

## Data Availability

The data presented in this study are available upon request from the corresponding author.

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
