# Peer review of "Design Optimization of Lattice Structures under Compression: Study of Unit Cell Types and Cell Arrangements"

_materials, 2021, doi:10.3390/ma15010097_

Round 1
Reviewer 1 Report
Please see the attachment.

Reviewer 2 Report
The paper proposes an approach to reach the optimized geometry of the additive manufactured unit cell of lattice structures. A parametric CAD model was realized, and an optimization process based on the results of finite element analysis is applied. Comparative results of the considered lattice geometries are considered. Some specimens were also manufactured and tested to evaluate the compressive behavior.
The manuscript is clear and well-written, and the methodology is properly exposed. The conclusions are clearly reported. Some issues arise in the description of the FEA method and in the outline of the results. Consequently, the following aspects should be considered in the revised version of the manuscript:
- Page 7 - line 191, the yield strength is an intrinsic property of the material, it does not depend on the structure geometry.
- Page 7 – line 199, the provided material properties are typical of the continuum material. Considering a 3D printed object, it is reasonable to expect a reduction of both the elastic and the strength properties. Did the authors consider these aspects?
- Section 2.2.1, the description of the mesh is not sufficient. Was the mesh mapped or free? How many elements does the mesh contain? Did the authors perform a convergence analysis? Explain the application strategy of the external load. A figure may help the description.
- Section 2.2.1 line 207-214, this segment of the section is unclear. How did the authors measure the stress? Are they referring to an apparent stress on the equivalent cross-section of the lattice structure or to a nodal/element solution? Even in this case, a contour map figure would be useful.
- According to Figure 8 and Figure 9, it would be interesting to further show the specific yield force (force per unit volume) in order to show which design solution presents the best structural performance. In fact, it would also be interesting to compare the unit cells geometry considering lattice structure domains with the same mass (and consequently with different domain sizes).
- The scientific literature about the lattice structures is considerably wide and other references should be added:
- https://doi.org/10.1016/j.euromechsol.2021.104291
- https://doi.org/10.1016/j.ijsolstr.2018.10.005
Reviewer 3 Report
This paper presents an interesting research idea, supported by simulation and experimental studies. However, this paper is missing some critical information, which can undermine the contribution of this research. Authors need to address the following comments.
- The limitation of the proposed design process/database needs to be discussed. For example, can it be used for designing larger sized parts, say 200mm*200mm*200mm? The paper only investigated one dimension, it is not straightforward to conclude the applicability of the proposed method.
- How did the authors choose the three relative densities (0.1, 0.2, and 0.3)?
- Different types of unit cells and geometrical designs can be leveraged to achieve different properties. The authors need to justify why and how can the design optimization method be applicable to optimize different unit cell types of geometries.
- For the numerical simulation, what is the mesh element size? Also, did the authors do a convergence study to check if the solution is element independent? Why did the authors choose the parabolic tetrahedral solid elements? What is the mesh density?
- A list of boundary conditions used in the simulation needs to be added.
- In the experiments, what is the selected layer thickness? What is the averaged build time for each part? How were the parts built, one part at a time or batch production?
- As stated in the paper, the yield force decreased when moved from the 3*3*3 system to the 4*4*4 system, please explain why.
- The print quality shown in Figure 11 is not satisfactory, which could affect the compressive test results. It looks like those parts need to be better post-processed. A closer look at the printed parts and an examination on their quality are needed.
- There is a huge difference between linear static FEA and the experiment test though nonlinear FEA improves the result. It looks like something is missing from the linear static FEA analysis.
- The authors have made some bold statements and conclusions, which are too general and can’t be valid based on the studies conducted in this research. It is suggested that authors check their entire manuscript and revise.
Reviewer 4 Report
This manuscript studies the effect of unit cell topology, pattern arrangement and relative density on the mechanical properties of lattice structures. After evaluating 11 types of unit cells and different lattice patterns by FEA and experimental tests, this study found out that 3X3X3 simple cubic lattice shows the best compression strength. This study optimized the lattice structure configuration, unit cell array, and unit cell type. Overall, this manuscript is in high quality and the following comment should be considered before acceptance.
- In the section of optimizing the unit cell array, the author mentioned the Maxwell number. The unit cell with M<0 is bend-dominated and the unit cell with M≥0 is stretch-dominated. The stretch-dominated structures are stiff and strong. If so, the face-centered cubic, octahedron and octet-cross should be stiffer and stronger than the others. But the linear static analysis of the lattice structures in table 4 and figure 6 shows that the simple cubic structure is the strongest even M=-6. Please explain it more.
Round 2
Reviewer 2 Report
The manuscript has been revised according to my comments.
As far as I am concerned, the paper can be accepted for publication.
Author Response
Round 2 response to Reviewer 2
The manuscript has been revised according to my comments.
As far as I am concerned, the paper can be accepted for publication.
Response: We would like to thank Reviewer 2 for the time and effort in reviewing our manuscript and providing positive feedback. Your previous comments helped us improve the manuscript and produce a more balanced and better account of the research. We are happy to know that our revisions have addressed all your concerns.
This is a basic study to determine the effects of the behavior of lattice structures on their description and prediction by identifying the correlation between unit cell types, lattice structure topologies, relative densities, unit cell arrangement patterns, and mechanical properties. We will be continuing our research on lattice structures and will report our findings in this journal.
Reviewer 3 Report
The authors have addressed some of my comments, but left some other comments unanswered.
- As the authors stated, this paper is limited to 20mm*20mm*20mm. So are the results scalable? How are these results useful to AM or metamaterial designers considering this limitation?
- The selection of relative density of 0.1, 0.2 and 0.3 does not make sense. Please provide proof from literature or justification.
- How is the optimization method presented in this research applicable to optimizing different unit cell types?
- The SLM fabricated parts demonstrate defects, showing the need for improving the selection of process parameters. Without doing that, what is the point of even optimizing metamaterial design? When choosing different printing process parameters, the optimal metamaterial design parameters could be significantly different. The authors first need to ensure a good fabrication quality. I suggest the authors redo the experiments.
Author Response
Round 2 response to Reviewer 3
The authors have addressed some of my comments, but left some other comments unanswered.
1.As the authors stated, this paper is limited to 20mm*20mm*20mm. So are the results scalable? How are these results useful to AM or metamaterial designers considering this limitation?
Response: We would like to thank Reviewer 3 for the time and effort in reviewing our manuscript and providing comments and suggestions, which have considerably helped us improve our manuscript. We apologize for any previous unanswered comments. In this round, we have answered each of your points below and hope that our responses and revisions address all your comments.
To evaluate material properties, test specimens are generally manufactured and tested according to certain standards. Using these test results, the mechanical properties of large structures are evaluated. For example, the overall structural strength of concrete is evaluated in terms of the compressive strength of a concrete cylinder specimen fabricated and tested according to a standard.
The standard related to the compressive strength of metal lattice structures is ISO 13314:2011 (Mechanical Testing of Metals — Ductility Testing — Compression Test for Porous and Cellular Metals.). According to ISO 13314, all spatial dimensions of the compression specimen shall be no less than 10 mm, with a sample-length-to-edge-length ratio of between 1 and 2. Currently, the standard does not provide an exact size; thus, the dimensions of the compressive strength specimens in related studies are different (see the table below).
|
Compression specimen dimensions |
Doi [ref.] |
|
20 mm × 20 mm × 20 mm |
https://doi.org/10.3390/ma14092462 [75] |
|
20 mm × 20 mm × 20 mm |
https://doi.org/10.1016/j.actamat.2018.08.030 [76] |
|
20 mm × 20 mm × 20 mm |
https://doi.org/10.1016/j.ijmecsci.2021.106922 [77] |
|
18 mm × 18 mm × 18 mm |
https://doi.org/10.1016/j.addma.2017.04.003 |
|
18 mm × 18 mm × 18 mm |
https://doi.org/10.1016/j.ijmecsci.2020.105480 |
|
14.4 mm × 14.4 mm × 14.4 mm |
https://doi.org/10.3390/ma14205962 |
|
24 mm × 24 mm × 24 mm |
https://doi.org/10.3390/met10020213 |
We believe that research on the lattice structure is currently in its early stages. However, many studies have been conducted to analyze the mechanical properties of lattice structures. With these accumulating studies, we believe that it is possible to apply the lattice structure to a real structure.
Therefore, we have revised the relevant paragraph in Subsection 2.2.1 and the Conclusions section.
[Subsection 2.2.1]
Numerical modeling of the optimal arrangement of unit cells for cubic lattice structure was performed to investigate the axial compressive behavior. The dimensions of the cubic lattice structure were designed to be 20 mm × 20 mm × 20 mm, referring to the standard ISO 13314 [74] and related studies [75-77].
[Conclusions section]
This study was conducted using a limited structure size (dimensions of 20 mm × 20 mm × 20 mm). Therefore, it is essential to analyze and optimize the actual size to be manufactured. Moreover, further investigation is necessary to confirm this behavior in parts with larger overall dimensions. In addition, the unit cell types, lattice structure topologies, relative densities, and unit cell array patterns should be determined with respect to the actual structural requirements (such as compression member, bending member, and energy absorbing member).
- The selection of relative density of 0.1, 0.2 and 0.3 does not make sense. Please provide proof from literature or justification.
Response: Thank you for your suggestion.
Generally, the relative density of the lattice structure is defined as follows:
The mechanical properties of lattice structures can be predicted as a function of their unit cell geometries, relative densities, sizes, strut dimensions, and arrangement. In particular, for lattice structures, relative density, which is the ratio of the apparent density of the lattice structure to the density of the material with a solid structure, dictates its mechanical properties [25-28]. (see the revised manuscript, Lines 142–146, and the Figure below)
Therefore, in this study, the relative density was kept constant to compare the mechanical performance of each unit cell lattice structure (see Figure 10).
---------------------------------------------------------------------------------------------------------------------------------------
[Figure for explanation]
The lattice structures are generally aimed at a high strength-to-relative-density ratio. For this reason, many studies related to lattice structures employ a low relative density (see the table below).
|
Relative density (percentage) |
doi |
|
0.07 to 0.42 (7% to 42%) |
https://www.mdpi.com/1996-1944/13/9/2204/htm |
|
0.05 to 0.30 (5% to 30%) |
https://www.mdpi.com/1996-1944/14/9/2462/htm |
|
0.13 to 0.41 (13% to 41%) |
https://doi.org/10.1016/j.matdes.2018.11.035 |
|
0.092 to 0.191 (9.12% to 19.14%) |
https://doi.org/10.1016/j.matdes.2021.110140 |
|
0.035 to 0.138 (3.5% to 13.8%) |
https://doi.org/10.1016/j.ijmecsci.2013.01.006 |
|
0.053 to 0.166 (5.3% to 16.6%) |
https://doi.org/10.1016/j.ijimpeng.2007.10.005 |
|
0.07 to 0.222 (0.7% to 22.2%) |
https://doi.org/10.1016/j.matdes.2019.107685 |
|
0.075 to 0.397 (7.5% to 39.7%) |
https://doi.org/10.1016/B978-0-08-100433-3.00005-1 |
|
* In general, the relative density in lattice structures is not expressed in [%] but rather as a relative density of 0–1 [no units]). In addition, most of the currently reported unit cell designs set the strut length to an integer value, so the relative density is not an integer. In this study, r/s data ​​were provided to control the relative density. |
|
Therefore, in this study, relative density values ​​were set to 0.1, 0.2, and 0.3, considering the light-weight structure.
Thank you for re-considering that these values (relative densities of 0.1, 0.2 and 0.3) were set for to accumulate more data.
Therefore, we have revised the relevant paragraph in Subsection 2.2.1 (line 184–185)
their relative densities were designed as 0.1, 0.2, and 0.3, respectively, considering the light-weight structure.
- How is the optimization method presented in this research applicable to optimizing different unit cell types?
Response: Using the results of this study, the structure members can be optimized for other uses. For design purposes, we provided r/s values to control the relative densities of the unit cells.
For example, when designing a flexural lattice structure with a relative density of 0.3 (=70% weight reduction), a unit cell with a relative density of 0.3 is placed on the flexural member. By controlling the unit cell type and cell arrangement, an optimal lattice structure can be designed. We think this optimization can be verified through FEA.
Revised manuscript, Subsection 3.1, lines 261–264: “The relative densities and r/s values of the 11 types of unit cells (relative density in the range of 0.005–0.900) are shown in Table 3 and Figure 6. By setting r/s as a variable, it was possible to control the relative densities of the unit cells.”
Revised manuscript, Conclusions section: “This study was conducted using a limited structure size (dimensions of 20 mm × 20 mm × 20 mm). Therefore, it is essential to analyze and optimize the actual size to be manufactured. Moreover, further investigation is necessary to confirm this behavior in parts with larger overall dimensions. In addition, the unit cell types, lattice structure topologies, relative densities, and unit cell array patterns should be determined with respect to the actual structural requirements (such as compression member, bending member, and energy absorbing member).”
- The SLM fabricated parts demonstrate defects, showing the need for improving the selection of process parameters. Without doing that, what is the point of even optimizing metamaterial design? When choosing different printing process parameters, the optimal metamaterial design parameters could be significantly different. The authors first need to ensure a good fabrication quality. I suggest the authors redo the experiments.
Response: In this study, the equipment output parameters were fixed owing to time and budget limitations. Metal AM is very expensive, so the number of specimens that can be manufactured is limited. Based on our experience, the output parameters are at a level that allows for high-quality manufacturing. However, we recognize the need to diversify the equipment output parameters. Thus, we plan to continue this research in the future.
Revised manuscript, Subsection 3.1, lines 246–251: “The thickness of the layer was 0.02 mm, and the following parameters were applied: laser power of 185 W, scanning speed of 1100 mm∙s-1, and hatching distance of 90 µ; these parameters were kept constant for all specimens. The building volume of the manufacturing equipment was 250 mm × 250 mm × 250 mm. The specimens were placed on the same layer of the building volume and manufactured in batches, and the manufacturing time was approximately 20 h.”
Revised manuscript, Subsection 3.1, lines 194–202. “The values of the material properties of the titanium alloy (Ti-6Al-4V) used in this study were derived experimentally. The specimen was manufactured following the ASTM B988-18 standard [78], and the manufacturing parameters were the same as those mentioned in Subsection 2.3.2. In terms of the results, an elastic modulus of 119.0 GPa, yield strength of 1 125.0 MPa, tensile strength of 1 200.0 MPa, and Poisson's ratio of 0.34 were achieved. The derived mechanical properties satisfied the ASTM B988-18 standard (yield strength ≥ 828 MPa and ultimate tensile strength ≥ 895 MPa). The aforementioned material property values were used for the numerical analysis.”
Regarding the test specimen quality, residues seem to be left behind in the simple cubic structure in the post-processing process (Figure 12). In accordance with your suggestions, we re-manufactured this structure under the same equipment output parameters. In addition, we keep the post-processing as clean as possible. Next, the compressive strength was tested again. The re-tested maximum yield force was 86 300 N, which is 97.7% of the previous test value, confirming that the difference in the test values ​​was not large. Because the post-processing residue involves weak curing, the compression strength results were not significantly different.
Residues from post-processing Specimen for re-testing
Modified Figures 12 and 13
Figure 12. Compressive deformation response of lattice structures with relative density of 0.3 and 3 × 3 × 3 pattern (simple cubic, truncated cube, truncated octahedron, and octahedron).
Figure 13. Force–strain response of lattice structures at a relative density of 0.3 and for the 3 × 3 × 3 pattern (simple cubic, truncated cube, truncated octahedron, and octahedron).
